# Analytical Study and Comparison of Electromagnetic Characteristics of 8-Pole 9-Slot and 8-Pole 12-Slot Permanent Magnet Synchronous Machines Considering Rotor Eccentricity

**Hoon-Ki Lee, Tae-Kyoung Bang** **, Jeong-In Lee, Jong-Hyeon Woo** **, Hyo-Seob Shin** **, Ick-Jae Yoon** **and Jang-Young Choi ***

Department of Electrical Engineering, ChungNam National University, Daejeon 34134, Korea;
lhk1109@cnu.ac.kr (H.-K.L.); bangtk77@cnu.ac.kr (T.-K.B.); lji477@cnu.ac.kr (J.-I.L.); dnwhd0@cnu.ac.kr (J.-H.W.);
shs1027@cnu.ac.kr (H.-S.S.); ijyoon@cnu.ac.kr (I.-J.Y.)
* Correspondence: choi_jy@cnu.ac.kr; Tel.: +82-042-821-7601

**Abstract:** In this study, a magnetic field is analyzed using an analytical method and compared with the electromagnetic characteristics of 8-pole 9-slot and 8-pole 12-slot permanent magnet synchronous machines considering rotor eccentricity. The magnetic flux density and back electromotive force (EMF) are derived using perturbation theory and electromagnetic theory. First, the Fourier modeling of a permanent magnet is performed through magnetization modeling, and two analysis regions are set based on several assumptions for applying the analytical method. Accordingly, the governing equations of the analysis regions are derived in the form of Poisson and Laplace equations. In addition, the undefined coefficients of the general solutions are calculated through general solutions and appropriate boundary conditions, and the magnetic flux density and back EMF of the air gap region are derived based on the definition of the magnetic vector potential. The results obtained using the analytical method are compared with the finite element method and experimental results. In addition, we perform a torque analysis considering rotor eccentricity and analyze the torque ripple based on rotor eccentricity for two cases involving the pole/slot combination.

**Keywords:** PMSM; analytical method; perturbation theory; torque; torque ripple; rotor eccentricity

## 1. Introduction

Permanent magnet synchronous machines (PMSMs) are becoming popular as a key technology for applications such as home appliances, industrial tools, and electric vehicles because of their high efficiency, high power density, and low maintenance cost; in addition, interest in resolving motor malfunction has increased. Data provided in [1–3] indicate that 41% of motor faults are bearing faults, 37% are stator faults, 10% are rotor faults, and 12% are other faults. One of the main causes of malfunction is rotor eccentricity, wherein the center of the rotor axis deviates from the center of the stator, resulting in a non-uniform air gap. Static eccentricity, which is a type of rotor eccentricity, is a condition where the position of the minimum radial air gap is fixed [2–5]. This can be caused by stator core ovality, incorrect positioning of the stator core, or bearing at commissioning or following a repair, and its level does not change over time. Furthermore, the magnetic flux density in the air gap is an important characteristic of the machine performance. Therefore, the impact of rotor eccentricity on the magnetic field distribution must be analyzed for predicting the characteristics of PMSMs [6,7]. The finite element method (FEM) and analytical methods are employed to design PMSMs or perform characteristic analysis. The FEM is a numerical process that can be performed using a commercial tool [8]. However, it is necessary to learn to use each commercial tool, which is disadvantageous as the experience of the designer is necessitated for an accurate analysis. In the analytical method, electromagnetic field characteristics are analyzed using Maxwell's equation. Magnetization modeling

is applied to the design of the machine using the Fourier series and partial differential equations derived using the magnetic vector potential. It must be preceded by deriving a solution that considers various boundary conditions. Several studies are being conducted because the analytical method can rapidly predict the characteristic variation based on the design parameters [9].

The magnetic field is analyzed using an analytical method and compared with the electromagnetic characteristics of 8-pole 9-slot and 8-pole 12-slot PMSMs considering rotor eccentricity. The magnetic flux density and back-electromotive force (EMF) are derived using perturbation theory and electromagnetic theory. The results obtained using the analytical method are compared with the FEM and experimental results. In addition, we performed a torque analysis based on rotor eccentricity and analyzed the torque ripple based on rotor eccentricity for two cases involving the pole/slot combination.

## 2. Electromagnetic Analysis of a PMSM Using Analytical Method

### 2.1. Analytical Model

Figure 1 shows the analysis models of 8-pole 9-slot and 8-pole 12-slot PMSMs. First, the analysis regions are defined for applying the analytical method, and the governing equations are derived using electromagnetic theory regarding each region. The magnetic flux density can be derived through the general solutions of the defined governing equations and the boundary conditions of each region. To apply the analytical method, the analysis model is simplified to the slotless model, the relative permeability of the iron core is infinite, the relative permeability of the permanent magnets (PMs) is equal to that of air, and the eddy effect is disregarded to apply the analytical method. Figure 2 shows the simplified model for applying the analytical method. The analysis regions are classified as an air gap region (Region I) and PM region (Region II), where rotor eccentricity is determined based on the stator coordinate $(r,\theta)$, and rotor coordinate $(\xi,\psi)$. $O_s$ and $O_r$ are the centers of the stator and rotor, respectively; $R_s$, $R_m$, and $R_r$ are the radii of the stator inner, PM surface, and rotor inner, respectively. The relationship between the parameters of the $r$-$\theta$ coordinate and $\xi$-$\psi$ coordinate is organized based on Taylor' expansion, as follows [9].

$$\begin{aligned} \xi &= r - \varepsilon \cos(\theta - \phi) + O(\varepsilon^2) \\ \psi &= (\theta - wt) + \tfrac{\varepsilon}{r}\sin(\theta - \phi) + O(\varepsilon^2), \qquad \varepsilon = ec \times g \end{aligned} \tag{1}$$

where $r$ is the length from the stator center to the observation point, $\varepsilon$ is the length of rotor eccentricity, $ec$ is the eccentricity ratio, and $g$ is the nominal air gap length. $O(\varepsilon^2)$ is the second order of the perturbation term. Only the first-order term of the perturbation term is considered, and the second-order term is disregarded in this study, since rotor eccentricity is usually small in comparison to the air gap length.

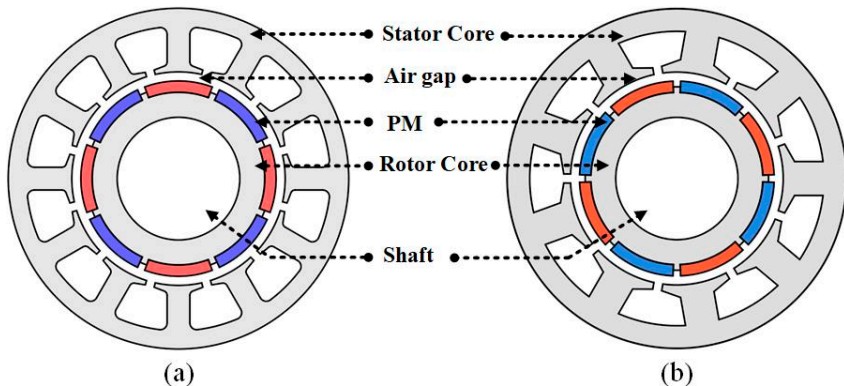

**Figure 1.** Analysis model: (**a**) 8 pole, 12 slot; (**b**) 8 pole, 9 slot.

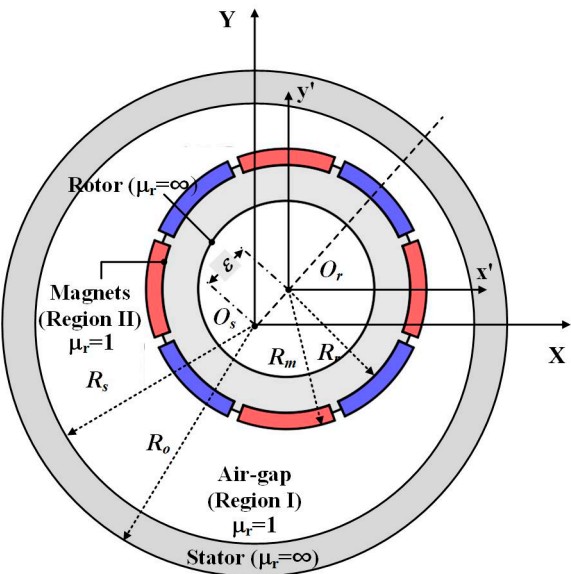

**Figure 2.** Simplified model for using analytical method.

### 2.2. Magnetic Field Analysis Using Analytical Method

To derive the magnetic flux density for PMs, Fourier modeling must be performed. Fourier modeling for PMs is expressed as follows:

$$\mathbf{M} = \sum_{n=1,3,5\cdots}^{\infty} M_{rn}\cos(np\theta)\mathbf{i}_r + M_{\theta n}\sin(np\theta)\mathbf{i}_\theta \tag{2}$$

where $\mathbf{M}$ is the magnetization vector, $M_{rn}$ and $M_{\theta n}$ are Fourier coefficients of the radial component and tangential component of the $r$- and $\theta$-direction, respectively. $n$ and $p$ are the space harmonic order and number of pole pairs, respectively. Rotor eccentricity can be treated as one of kind of perturbation phenomenon. Perturbation theory is necessitated to consider rotor eccentricity. Perturbation theory is a theory that expresses the solution of a problem that cannot be solved analytically as a Taylor series of parameters that can be considered extremely small. When this is applied to the magnetic vector potential, it is expressed as follows [9]:

$$\begin{aligned}
A_{z1}(r,\theta,\varepsilon) &= A_{z1}^{(0)}(r,\theta) + \varepsilon A_{z1}^{(1)}(r,\theta) + \varepsilon^2 A_{z1}^{(2)}(r,\theta) + \cdots \\
A_{z2}(r,\theta,\varepsilon) &= A_{z2}^{(0)}(r,\theta) + \varepsilon A_{z2}^{(1)}(r,\theta) + \varepsilon^2 A_{z2}^{(2)}(r,\theta) \cdots
\end{aligned} \tag{3}$$

where $A_{z1}$ and $A_{z2}$ are the magnetic vector potential of Region I and II, respectively. $A_{zn}^{(0)}$ and $A_{zn}^{(1)}$ are the zeroth and first orders of perturbation, respectively. $A_{zn}^{(0)}$ occurs when the rotor is not eccentric. $A_{zn}^{(1)}$ is generated by the rotor eccentricity effect without PMs or current excitation. The reason why even the first term is considered is that more accurate results can be derived when the second term is considered, but the calculation process becomes complicated, and reliable results can be obtained even when only the first term is considered. Therefore, the governing equation for each region is expressed as Poisson and the Laplace equations.

$$\nabla^2 \mathbf{A}_{z1}^{(0)} = \frac{\partial^2 A_{z1}^{(0)}}{\partial r^2} + \frac{1}{r}\frac{\partial A_{z1}^{(0)}}{\partial r} - \frac{q^2}{r^2}A_{z1}^{(0)} = 0 \tag{4}$$

$$\frac{\partial^2 A_{z2}^{(0)}}{\partial r^2} + \frac{1}{r}\frac{\partial A_{z2}^{(0)}}{\partial r} - \frac{q^2}{r^2}A_{z2}^{(0)} = -\mu_0 \nabla \times \mathbf{M} \tag{5}$$

$$\frac{\partial^2 A_{z1}^{(1)}}{\partial r^2} + \frac{1}{r}\frac{\partial A_{z1}^{(1)}}{\partial r} - \frac{q^2}{r^2}A_{z1}^{(1)} = 0 \tag{6}$$

$$\frac{\partial^2 A_{z2}^{(1)}}{\partial r^2} + \frac{1}{r}\frac{\partial A_{z2}^{(1)}}{\partial r} - \frac{q^2}{r^2}A_{z2}^{(1)} = 0 \tag{7}$$

The general solution of each region is derived from the Cauchy–Euler equation. The general solutions of each analysis region can be obtained as follows:

$$\mathbf{A}_{z1}^{(0)} = \sum_{n=1,3,5\cdots}^{\infty} \left(A_1 r^{np} + B_1 r^{-np}\right)\sin(np\theta) \quad \mathbf{i}_z \tag{8}$$

$$\mathbf{A}_{z2}^{(0)} = \sum_{n=1,3,5\cdots}^{\infty} \left(A_2 r^{np} + B_2 r^{-np} + \frac{\mu_0 rnp M_n}{(np)^2 - 1}\right)\sin(np\theta) \quad \mathbf{i}_z \tag{9}$$

$$\mathbf{A}_{z1}^{(1)} = \varepsilon \sum_{n=1,3,5\cdots}^{\infty} \begin{array}{l}\left(W_1 r^{np-1} + X_1 r^{-np+1}\right)\sin[(np-1)\theta + \phi] \\ +\left(Y_1 r^{np+1} + Z_1 r^{-np-1}\right)\sin[(np+1)\theta - \phi]\end{array} \quad \mathbf{i}_z \tag{10}$$

$$\mathbf{A}_{z2}^{(1)} = \varepsilon \sum_{n=1,3,5\cdots}^{\infty} \begin{array}{l}\left(W_2 r^{np-1} + X_2 r^{-np+1}\right)\sin[(np-1)\theta + \phi] \\ +\left(Y_2 r^{np+1} + Z_2 r^{-np-1}\right)\sin[(np+1)\theta - \phi]\end{array} \quad \mathbf{i}_z \tag{11}$$

The effect of the rotor eccentricity can be observed at the rotor yoke surface and the interface between the PMs and air gap in the $r$-$\theta$ coordinate. In the $\xi$-$\psi$ coordinate, the radius corresponding to the surface and interface are $R_r$ and $R_m$, respectively. By applying $\rho = R_r$ and $R_m$ in Equation (1), the following equations can be obtained:

$$f_{R_r}(r,\theta,\varepsilon) = r - \varepsilon\cos(\theta - \phi) - R_r \tag{12}$$

$$f_{R_m}(r,\theta,\varepsilon) = r - \varepsilon\cos(\theta - \phi) - R_m \tag{13}$$

The normal direction vector of the rotor core surface and PM surface can be derived by applying the gradient ($\nabla$) to Equations (12) and (13).

$$\mathbf{n}_{R_m} = \nabla f_{R_m} = \mathbf{i}_r + \frac{\varepsilon}{r}\sin(\theta - \phi)\mathbf{i}_\theta \tag{14}$$

$$\mathbf{n}_{R_s} = \nabla f_{R_s} = \mathbf{i}_r + \frac{\varepsilon}{r}\sin(\theta - \phi)\mathbf{i}_\theta \tag{15}$$

The boundary conditions can be derived using Equations (14) and (15) and electromagnetic theory. The boundary conditions at the rotor core surface and PM–air gap interface can be expressed as follows:

$$\mathbf{n}_{R_r} \times \mathbf{H}_2 = 0 \tag{16}$$

$$\mathbf{n}_{R_m} \times (\mathbf{H}_1 - \mathbf{H}_2) = 0 \tag{17}$$

$$\mathbf{n}_{R_m} \cdot (\mathbf{B}_1 - \mathbf{B}_2) = 0 \tag{18}$$

The boundary conditions of the zeroth and first order can be derived by applying $\nabla \times \mathbf{A} = \mathbf{B}$ to Equation (3) and substituting it into Equations (16)–(18). $A_1$, $B_1$, $A_2$, $B_2$, $W_1$, $X_1$, $Y_1$, $Z_1$, $W_2$, $X_2$, $Y_2$, and $Z_2$ of Equations (8)–(11) can be calculated using the derived boundary conditions. The magnetic flux density of the $r$- and $\theta$-directions can be express as follows [6]:

$$\mathbf{B}_r = \frac{1}{r}\frac{\partial \mathbf{A}}{\partial \theta}, \mathbf{B}_\theta = -\frac{\partial \mathbf{A}}{\partial r} \tag{19}$$

The zeroth order of the perturbation term implies no rotor eccentricity, and the first-order term refers to the amount of change in magnetic flux density based on rotor eccentricity; therefore, the final magnetic flux density is the sum of the zeroth- and first-order magnetic flux density. The magnetic flux density considering rotor eccentricity can be expressed as:

$$\mathbf{B} = \mathbf{B}^{(0)} + \varepsilon\mathbf{B}^{(1)} \tag{20}$$

### 2.3. Back-EMF

To obtain the back-EMF, the magnetic flux linking the coil must be determined as follows [10]:

$$\Phi_{coil,(1,\cdots,Q)} = \int_{-\theta_c/2}^{\theta_c/2} \mathbf{B} \cdot d\mathbf{S} \quad , \qquad E(t) = -N_c \frac{d\left(\Phi_1 + \cdots + \Phi_Q\right)}{dt} \qquad (21)$$

where $\mathbf{B}$ is the magnetic flux density vector in the stator surface region, and $d\mathbf{S}$ is an element of the coil surface area vector. $Q$ is the slot number/phase. $\theta_c$ denotes the mechanical angle per slot. The back-EMF can be obtained using Faraday's law. $N_c$ is the number of turns in each coil. Figure 3 is an example explaining the calculation of the back-EMF of the 8-pole 9-slot. The back-EMFs in other coils of the same phase are not necessarily similar because of the eccentricity. They must be calculated individually by performing an appropriate shift.

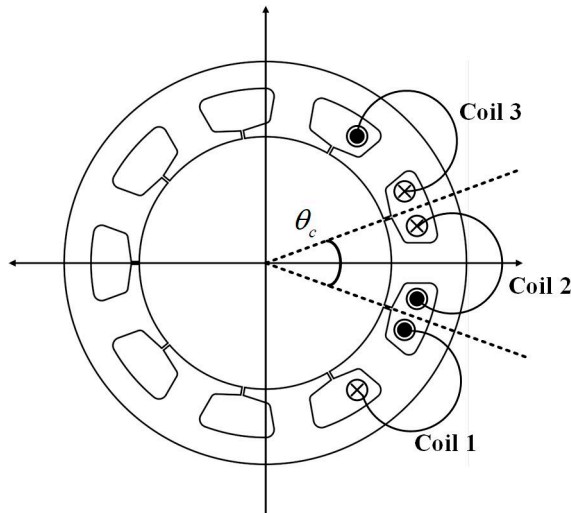

**Figure 3.** Winding distribution of 8-pole 9-slot.

## 3. Results and Discussion

Figure 4 shows the experimental set. Table 2 shows the design specifications. The stator and rotor were made of 50PN470, whereas the permanent magnet was made of N42SH; the experiment was performed at room temperature. The servo motor was installed on the opposite side of the test motor to measure the back-EMF. Figure 5a shows the magnetic flux density in the air gap when no eccentricity occurred, and Figure 5b is the variation in the magnetic flux density according to rotor eccentricity. Figure 6 shows the magnetic flux density in the air gap when 25% rotor eccentricity occurred. A 25% eccentricity is based on nominal air gap length. Since the air gap of the analysis model is 4 mm, it means that 1 mm eccentricity occurs. The analytical results were verified by comparing them with the FEM results. FEM analysis was performed using ANSYS Electronics 2020R2. It was confirmed that the results agreed well. Figure 7 is the mesh plot of non-simplified analysis models. Non-simplified model in Figures 8 and 9 are the slotted models. FEM analysis and experiments are performed using slotted models. Figure 8a,b show the back-EMFs of the 8-pole 9-slot PMSM when no rotor eccentricity and 25% rotor eccentricity occurred, respectively. The analytical results of the back-EMFs of the 8-pole 9-slot PMSM were compared with the FEM and experimental results. It was observed that the results exhibited good agreement with each other. In the 8-pole 9-slot model, the winding arrangement distributed asymmetrically to one side, as shown in Figure 8b; therefore, if eccentricity occurs, the result of the unbalanced back-EMF can be confirmed. Figure 9a,b show the back-EMF of the 8-pole 12-slot PMSM based on rotor eccentricity. As shown in Figure 9b, the winding arrangement of the 12-slot stator was distributed symmetrically. Consequently, the effect of rotor eccentricity was nullified, and the results of

the cases with and without rotor eccentricity were compared. Tables 1 and 3 summarize the analysis results. The error of the analytical results is confirmed to be within 5%. Figure 10a, b show the results of the output torque and torque ripple analysis based on the eccentricity of the 8-pole 9-slot and 8-pole 12-slot PMSMs. When comparing those PMSMs, it was observed that the cogging torque of the 8-pole 12-slot PMSM was larger than that of the 8-pole 9-slot PMSM; therefore, the torque ripple ratio of the 8-pole 12-slot PMSM was larger than that of the 8-pole 9-slot under a low input current. However, in the 8-pole 9-slot model, the increasing rate of the torque ripple based on eccentricity was larger than that of the 8-pole 12-slot owing to the asymmetric winding arrangement. Under the low output power condition, the 8-pole 9-slot PMSM indicated a sinusoidal back-EMF, resulting in a small torque ripple ratio in the output torque. However, as the output power increased, the 8-pole 12-slot PMSM was less affected by the rotor eccentricity.

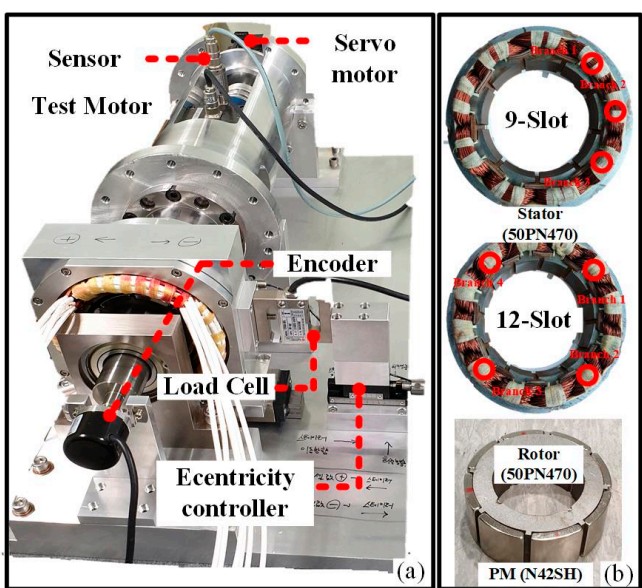

**Figure 4.** Experimental set: (**a**) experiment configuration; (**b**) manufactured model.

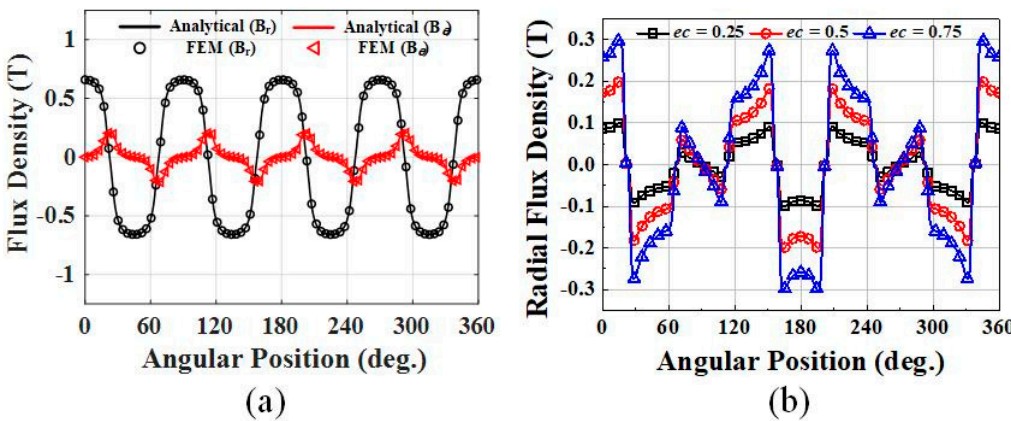

**Figure 5.** Flux density: (**a**) without rotor eccentricity; (**b**) variation in radial flux density according to rotor eccentricity.

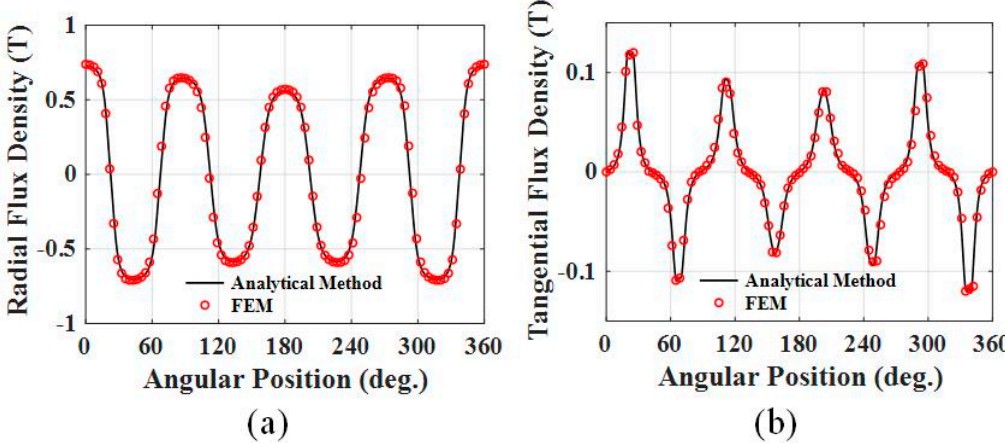

**Figure 6.** Flux density with 25% rotor eccentricity at the air gap: (**a**) radial flux density; (**b**) tangential flux density.

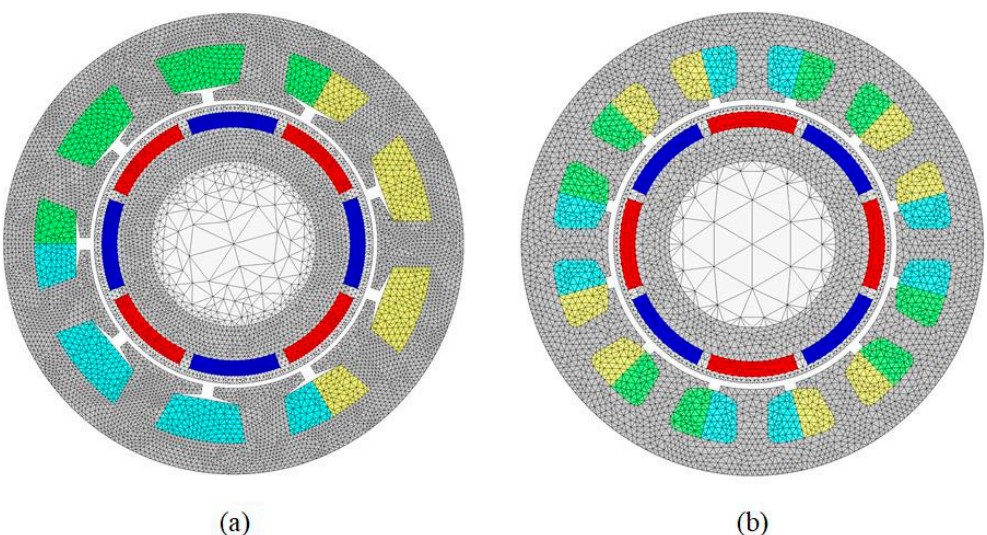

**Figure 7.** Mesh plot of analysis models: (**a**) 8−pole 9−slot; (**b**) 8−pole 12−slot.

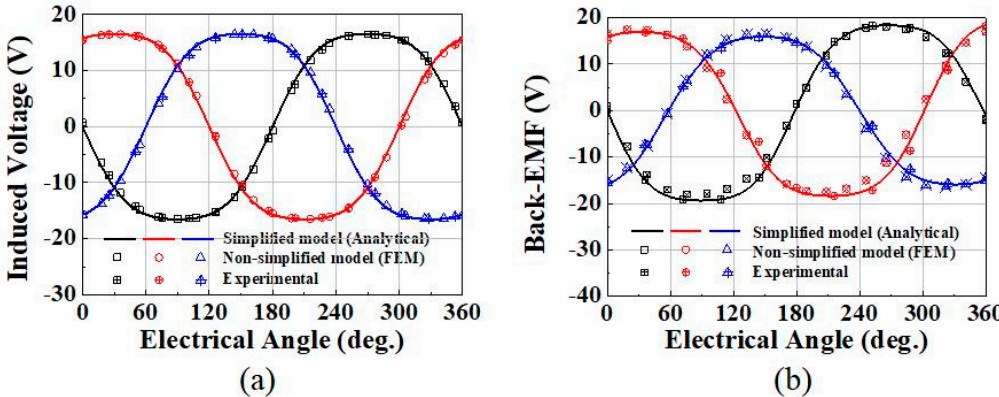

**Figure 8.** Back−EMF of 8−pole 9−slot PMSM: (**a**) without rotor eccentricity; (**b**) with 25% eccentricity.

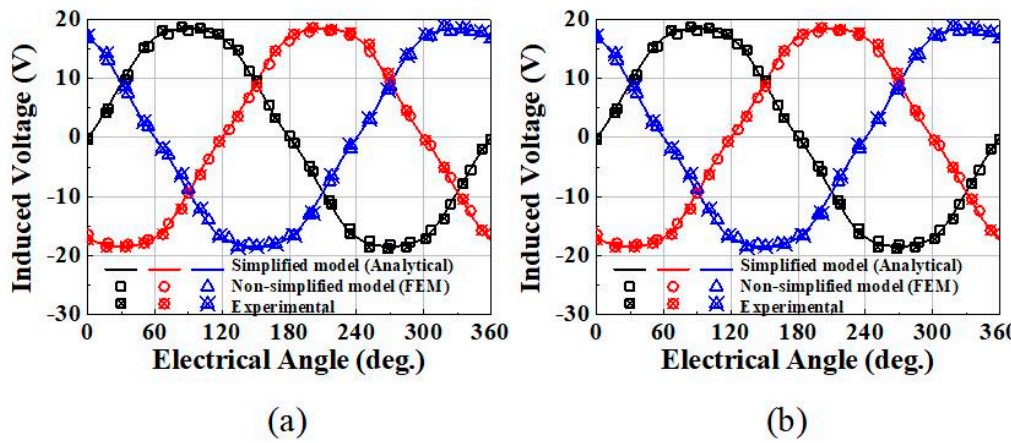

**Figure 9.** Back-EMF of 8−pole 12-slot PMSM: (**a**) without rotor eccentricity; (**b**) with 25% eccentricity.

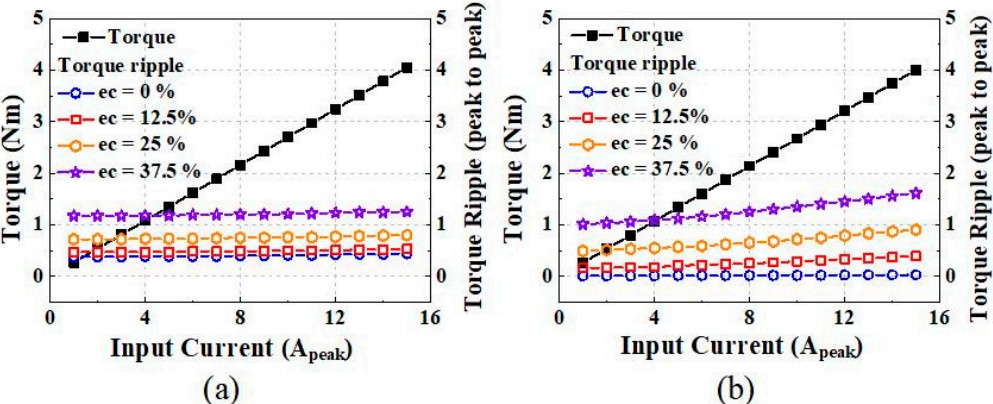

**Figure 10.** Torque characteristics: (**a**) 8−pole 12−slot and (**b**) 8−pole 9−slot.

**Table 1.** Analysis results of back-EMF without rotor eccentricity.

| Parameter | Analytical | | FEM | | Measurement | |
|---|---|---|---|---|---|---|
| | 8p9s | 8p12s | 8p9s | 8p12s | 8p9s | 8p12s |
| Number of meshes | - | - | 31,496 | 30,541 | - | - |
| Analysis time (s) | 0.11 | 0.12 | 11 | 23 | - | - |
| Back-EMF ($V_{max}$) | 17.63 | 18.45 | 17.52 | 18.35 | 17.68 | 19.05 |
| Error (%) | 0.28 | 3.14 | 0.9 | 3.67 | - | - |

**Table 2.** Design specification of analysis model.

| Parameter | Value | Unit |
|---|---|---|
| Number of slots/poles | 9/8, 12/8 | mm |
| Outer radius of stator | 75 | mm |
| Inner radius of stator | 47 | mm |
| Outer radius of rotor | 43 | mm |
| Thickness of PMs | 5 | mm |
| Axial length | 30 | mm |
| Magnet remanence | 1.27 | T |
| Rated speed | 1000 | Rpm |
| Pole arc ratio | 0.9 | |

**Table 3.** Analysis results of back-EMF with 25% rotor eccentricity.

| Parameter | Analytical | | FEM | | Measurement | |
|---|---|---|---|---|---|---|
| | 8p9s | 8p12s | 8p9s | 8p12s | 8p9s | 8p12s |
| Number of meshes | - | - | 31,839 | 33,910 | - | - |
| Analysis time (s) | 0.11 | 0.12 | 12 | 25 | | |
| Back-EMF ($V_{max}$) | 18.36 | 18.45 | 18.2 | 18.35 | 18.6 | 19.04 |
| Error (%) | 1.29 | 3.1 | 2.15 | 3.62 | - | - |

## 4. Conclusions

In this study, we performed a magnetic field analysis based on rotor eccentricity using an analytical method, and the torque characteristics of a two pole/slot combination were compared. Based on perturbation theory and electromagnetic theory, the magnetic flux density and back-EMF in the air gap region were derived, and the analytical result was verified by comparing it with the FEM and the experimental results. Both models with slotted and simplified models without slots were analyzed using the FEM. We conformed that even if the magnetic flux density is predicted as slotless and the back-EMF is derived using it, a slotted model can obtain similar results. Subsequently, torque analysis was performed based on rotor eccentricity to analyze the variation in the torque ripple for the two pole/slot combinations. The analysis results from this study will benefit the design of motors.

**Author Contributions:** J.-Y.C.: conceptualization, review and editing; H.-K.L.: analysis, original draft preparation; T.-K.B.: magnetic field calculation; J.-I.L.: derivation of governing equation; J.-H.W.: experiment; H.-S.S.: torque analysis; I.-J.Y.: review and editing. All authors have read and agreed to the published version of the manuscript.

**Funding:** This work was supported by the Basic Research Laboratory (BRL) of the National Research Foundation (NRF-2020R1A4A2002021) funded by the Korean government.

**Data Availability Statement:** The data presented in this study are available on request from the corresponding author.

**Conflicts of Interest:** The authors declare no conflict of interest.

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
