# Peer review of "Analytical Study and Comparison of Electromagnetic Characteristics of 8-Pole 9-Slot and 8-Pole 12-Slot Permanent Magnet Synchronous Machines Considering Rotor Eccentricity"

_electronics, doi:10.3390/electronics10162036_

Round 1
Reviewer 1 Report
The paper addresses the magnetic field density (B) distribution and the induced back-EMF in 8-pole 9-slot and 8-pole 12-slot PMSM with rotor eccentricity. Under this scenario, two regions are fixed for each machine, which are governed by Poisson and Laplace equations. The formulation and boundary conditions are addressed with perturbation theory. In addition, the PM magnetization vector is represented by its Fourier series terms. Based on the magnetic potential vector, the B distribution is calculated, where the back-EMF is estimated with Faraday law. The results are compared to FEM analysis and experimental results.
General Overview:
The paper introduction needs to be more detailed: rotor eccentricity, which is a fundamental issue, is addressed in a very superficial way (e.g., static eccentricity is mentioned, but there is nothing about dynamic eccentricity). The description of the applied analytical method is too generic (for instance, the perturbation method is not discussed here); in addition, the state of the art is superficial.
Section 2 is the paper core. Similar to the introduction, a more thoroughly analysis and description are needed, starting with the perturbation method. In order to have a clear insight on these issues, I felt that reading reference [10] and [11] was fundamental.
As for Section 3, I believe Fig. 5 and 6 legends need a few corrections, while Fig. 7 and 8 need some clarification about the two FEM models (simplified and non-simplified). In addition, the discussion should have a deeper analysis, it can be improved.
In short, the paper does not give sufficient information about the method for reaching the governing and boundary equations, which is the basis of this work. In my opinion, it is not a “self-reading” paper, since references [10] and [11] are a fundamental backup. Moreover, the contribution given by the submitted paper is not too significant.
Please, address the following issues:
- Fig. 1 legend is not specific. Suggestion: “PMSM: 8 pole, 12 slot (a) and 8 pole, 9 slot (b)”.
- Eq. (2): Include “Magnetization vector” for M, as well as “radial component” and “tangential component” for Mrn and M_theta_n.
- Lines 94-95: More detail concerning perturbation theory should be provided here.
- Eq. (3): brackets are missing in several terms.
- Line 102: “Azn(1) is generated by the rotor eccentricity effect”. Thus, all the terms Azn(j), for j>1, are disregarded. It is important to justify this.
- Eq. (18): Parameter Q is the slot number/phase? E(t) is the back-EMF/phase? This should be clearly stated. Instead of Nph, I suggest using Nc, since this is the turn number of a single coil.
- Fig.5 and Fig. 6 address 9/8 or 12/8 configuration?
- Check Fig. 5b) legend: it should be “with rotor eccentricity”. Only the flux density radial component is represented, why?
- Fig. 6a) is also for rotor eccentricity of 25%? If it is the case, it seems that is not in agreement with Fig. 5b).
- Fig.7 mentions two FEM approaches (simplified model and non-simplified model), but there is no reference to this in Section 3. On the other hand, in Fig. 8 only the non-simplified model is referred. This is not clear, it is important to explain this issue.
- Lines 199-202: This is a significant and interesting result, which deserves a deeper analysis.
Author Response
Thank you for your comments

Reviewer 2 Report
The authors deal with the study of electromagnetic characteristics of permanent magnet synchronous machines. Here are some remarks:
- The novelty of the paper is low. The issue of EMF is well known and presented in the literature. What is the novelty of the paper?
- The results are shown on few charts. There are no data in the tables to verify the results.
- What is the measurement error? The authors should analyse the measurement error.
The paper should be rejected because of its low scientific value.
Author Response
Thank you for your comments

Reviewer 3 Report
The authors studied the eclectromagnetic characteristics of permanent magnet synchonous machines considering rotor eccentricity by analytical method. The analytical results agree with FEM results and experimental results. But there are still some problems.
(1) Fig 5-8 give some FEM results. Please give the governing equations, the boundary conditions by FEM are the same as that by analytical method.
(2) The object predicted by FEM is the model in Fig 1, or the simplified model in Fig.2?
(3) Fig. 3 shows that the analytical result agree we0ll with the experimental data. The experimental equipment in Fig.4 has slot, but the objects predicted by analytical method has no slot. Their structure are different, why the result is similar? Why the actual machine do not use the slotless structure?
(4) Section "3"results and discussion" compare the analytical result with ANSYS results andexperimental data, but do not give any comments about eccentricity and the design of permanent magnet synchonous machines.
(5) Section "4 conclusion" only tell us the research content, but there is not any interesting results.
Author Response
Thank you for your comments

Reviewer 4 Report
The paper considers an analytical study of two designs of Permanent Magnet Synchronous Machines. The subject of study is within the scope of the journal Electronics, focused among others on electrical circuits & devices.
PMSM have gained a lot of attention of electrical engineers working on machines and drives. The present paper is focused on an important aspect of their operation, which is usually not taken into account in the analyses, namely the effect of rotor eccentricity on torque and back-EMF. In most cases the analysis is limited to full symmetry of excitation and geometry. The present contribution fills the gap, thus it is very valuable for practitioners, since in real life cases there might be scenarios different from highly idealistic cases considered in literature.
The analytical expression for deviation between the stator and the rotor coordinate systems are derived from Taylor’s expansion limited to the first order term (lines 79-80). A sentence at the end of this paragraph (after line 81) might explain the readers why higher order terms are disregarded. The reviewer is aware that consideration of additional nonlinear terms, even the second order, might lead to a formidable complicacy of the model, however it should be explicitly written what are the possible maximum values of eccentricity, expressed e.g. in terms of the ratio d_rotor/d_stator when one can use the linear model.
The description of Figure 1 in the text (line 60) should be inverted, since the first model for analysis refers to a 8-pole 12-slot machine and the second to a 8-pole 9-lot one.
Figures 5-8 present comparisons of computations with the proposed method and FEM. However the text does not provide much information about the FEM method used (the reviewer means e.g. the software used (FEMM, ANSYS, FEMLAB?), a figure depicting mesh discretization of the machine parts might enrich the manuscript contents, some details on the number of finite elements, computational burden (type of computer used, time spent on computations etc.). In the present case the Readers may just observe a perfect agreement between two approaches. But a question arises, is this always true (the reviewer means here again that the authors should discuss the model assumptions in more detail).
Summing up: this is a very interesting paper, which should be published, however it needs a revision (a more thorough explanation).
Author Response
Thank you for your comments

Round 2
Reviewer 2 Report
The authors have improved their work according to reviewers remarks. The paper should be accepted in the present form.
Author Response
Thank you for your comments.

Reviewer 4 Report
The paper is better in the present version. The authors tried to address the issues raised in the reviews, however it is felt that yet anither revision might be necessary.
- The new tables appeared which include some information on the number of finite elements used. However just a sentence on the use of commercial FEM software (that it is Ansys) is not enough. The authors should devote some paper space to a more detailed description (which module and software version was used, were coupled (iterative) computations necessary, what was the computation burden (how long the computation lasted, on which machine etc.), as pointed earlier the Figure with exemplary meshing might be interesting.
- Which of the results the authors consider as reference: the results from computation using their approach or from FEM?
- The reviewer sees that the authors try to react to his remark on the applicability telling that in Figure 7 the results for eccentricity equal to zero and to 25% are shown. However the "eccentricity" remains a non defined term. If you use relative units, then it is obvious that you have some reference value, in this context it is natual to use the value for the air gap delta = 0.5(D-d), where D is the stator inner diameter and d is the rotor outer diameter. As pointed out previously, the value of the paper would be much improved if the authors tried to carry out some numerical tests to show at which values of deviation between the "gravity" centers of the rotor and the stator the discrepancies between the computed values from FEM and from the proposed computational model become larger e.g. than 5 or 10 %. This might be important for the designers of electrical machines.
- Summing up to this point the reviewer thinks this might be an interesting paper for the audience, yet the authors should still improve it. The paper is rather short in the present version. If it will be increased by a page or two with more detailed discussion, it will be even better and useful.
Author Response
Thank you for your comments

Round 3
Reviewer 4 Report
The paper after the second revision is ok now. It deserves publication in the present form.